# Neural Network Based on Multi-Scale Saliency Fusion for Traffic Signs Detection

**Haohao Zou, Huawei Zhan * and Linqing Zhang** 

School of Electronic and Electrical Engineering, Henan Normal University, Xinxiang 453007, China
* Correspondence: zhanhw@126.com

**Abstract:** Aiming at recognizing small-scale and complex traffic signs in the driving environment, a traffic sign detection algorithm YOLO-FAM based on YOLOv5 is proposed. Firstly, a new backbone network, ShuffleNet-v2, is used to reduce the algorithm's parameters, realize lightweight detection, and improve detection speed. Secondly, the Bidirectional Feature Pyramid Network (BiFPN) structure is introduced to capture multi-scale context information, so as to obtain more feature information and improve detection accuracy. Finally, location information is added to the channel attention using the Coordinated Attention (CA) mechanism, thus enhancing the feature expression. The experimental results show that compared with YOLOv5, the *mAP* value of this method increased by 2.27%. Our approach can be effectively applied to recognizing traffic signs in complex scenes. At road intersections, traffic planners can better plan traffic and avoid traffic jams.

**Keywords:** multi-scale context; traffic sign; attention; complex scenes; YOLOv5

## 1. Introduction

As artificial intelligence and transportation network technology continue to advance, traffic sign detection is in increasing demand in computer vision algorithms. In autonomous driving, the Advanced Driver Assistance System (ADAS) [1] has a significant effect. The ADAS system first collects the road environment during driving and identifies, detects, and tracks the data. Active safety technology that detects potential dangers as quickly as possible to attract the driver's attention and improve safety is also of vital importance. Two methods for detecting traffic signs in AI applications are color-based and shape-based [2–5]. For example, HIS, CIElab, HSL [6–8], etc. Traffic signs are essential for safety. However, their proportion in the image is small and low resolution, so they are not easily identified. Therefore, there are still some difficulties detecting traffic signs in practical application scenes.

Recently, target detection and recognition have been successfully applied using deep convolution neural networks [9–11]. The classic algorithm in the two-stage method, which is the R-CNN algorithm proposed by Ross Girshich in 2014 [12], uses the R-CNN model in the candidate regions to extract features and complete the classification in support vector machines (SVM) [13]. The model creates a bounding-box regression algorithm to calculate and test the coordinates of the candidate region. When the experimental results are compared, the average accuracy of the R-CNN algorithm is approximately 20% higher than that of the non-neural network algorithm.

Typical algorithms in the one-stage approach are yolo only look once (YOLO) and single shot multiBox detector (SSD) [14,15]. Wang et al. combined the YOLO network with the faster R-CNN [16] network and proposed the SSD target detection algorithm. The SSD algorithm generates object bounding boxes of different sizes over the whole image. It uses non-maximum suppression (NMS) [17] to combine highly overlapping bounding boxes into one bounding box, turn the candidate region into a regression problem, and locate the predicted box that is closest to the target. Thus, the calculation speed and accuracy are improved.

As scholars continue to improve algorithms, the performance of the YOLO algorithm series is gradually improving. Recent research [18] proposes a traffic sign detection network based on YOLOv1, which enhances detection speed and reduces hardware requirements. Another study [19] suggested a detection network based on YOLOv3, which improved the detection accuracy, but the real-time detection effect was not very good.

In the natural road environment, the detection of traffic signs is easily influenced by various factors. The traditional traffic sign detection algorithm has insufficient environmental adaptability, and the detection effect is inferior. Although many detection algorithms are highly accurate in the detection process, due to the lack of real-time detection performance, most algorithms can hardly be applied to practical detection tasks. Furthermore, because of the current detection algorithm's extensive framework and many parameters, it is difficult to deploy on the platform. Therefore, creating a target recognition algorithm with solid precision and fast response time is essential in the challenging environment where road signs are located.

Joseph Redmon proposed the YOLO algorithm, which processes images very quickly and is suitable for real-time processing. However, it has a poor detection effect for traffic signs and nearby objects, and the positioning is inaccurate. Although the current YOLOv5 algorithm has achieved good detection performance for traffic signs, it is easy to lose small target feature information in the feature extraction process. The network model is large and has much parameter information. It also provided the foundation for a single target detection algorithm, which possesses the properties of high adaptability and rapid detection speed. The YOLOv5 network selected in this paper is a lightweight version of YOLOv5, which is more in line with the requirements of real-time detection. The backbone network, the neck, and the head are the three sections of YOLOv5. The network structure of YOLOv5 is shown in Figure 1.

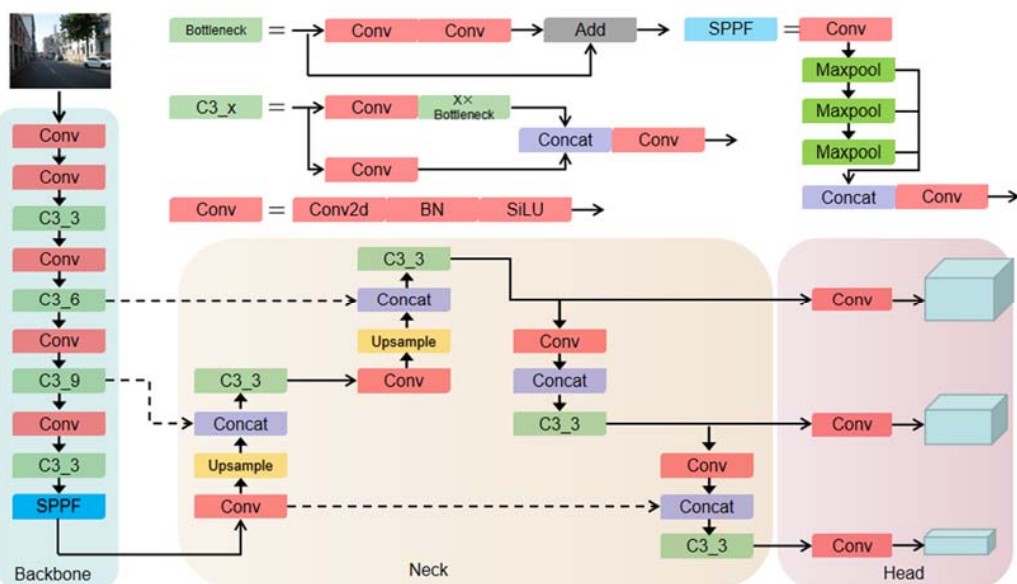

**Figure 1.** YOLOv5 network structure.

The YOLOv5 network draws on the design ideas of a cross-stage partial network (CSPNet), designs a C3 module containing multiple standard convolutional layers and multiple bottlenecks, and applies it to the backbone layer and the neck layer. Among them, the C3 module of the neck layer mainly learns the residual features while reducing the number of network parameters based on unchanged accuracy. The spatial pyramid pooling-fast (SPPF) module reduces the network layer based on the spatial pyramid pooling (SPP) module, removes redundant calculations, and performs feature fusion at a faster speed. In the neck layer, YOLOv5 uses feature pyramid network (FPN) + perceptual

adversarial network (PAN) for fusion. Among them, FPN fuses the obtained features in a top-down manner to bring predicted feature maps of various scales. The PAN layer consists of a bottom-up path added after the FPN layer. The two structures are combined, the features of the lower layer are passed up, and the parameters of the feature layer are aggregated from different backbone layers. Finally, the network makes predictions at the head layer.

In this paper, the YOLO-FAM algorithm is proposed, which improves the accuracy and speed of traffic sign recognition. The main contributions are as follows:

(1)　We combined the YOLOv5 network with ShuffleNet-v2, BIFPN, and CA mechanisms to propose the YOLO-FAM network, which solved the problem of traffic sign recognition in complex environments.

(2)　We conduct experimental evaluations to demonstrate the performance of the algorithm. Experimental results show that our algorithm performs close to optimality and outperforms many algorithms in realistic scenes.

This paper is organized as follows: Section 2 presents our approach. Section 3 presents the dataset setup. Section 4 presents the experimental results. Section 5 is dedicated to the conclusion.

## 2. Methodology

This paper proposes the YOLOv5-FAM algorithm. The YOLOv5-FAM algorithm uses the ShuffleNetv2 network instead of the original backbone network. It introduces the channel shuffle operation without increasing the amount of calculation and increases the effect of traffic sign feature extraction. It uses the BIFPN model instead of the original PANet model. It enhances the feature fusion and obtains more features. The CA attention mechanism is embedded into the feature fusion network, using the captured location information to capture the target area more accurately. The loss function of YOLOv5 hinders the model from effectively optimizing the similarity, so changing the loss function to EIOU loss makes convergence faster. Figure 2 depicts the improved YOLO-FAM network structure in this paper.

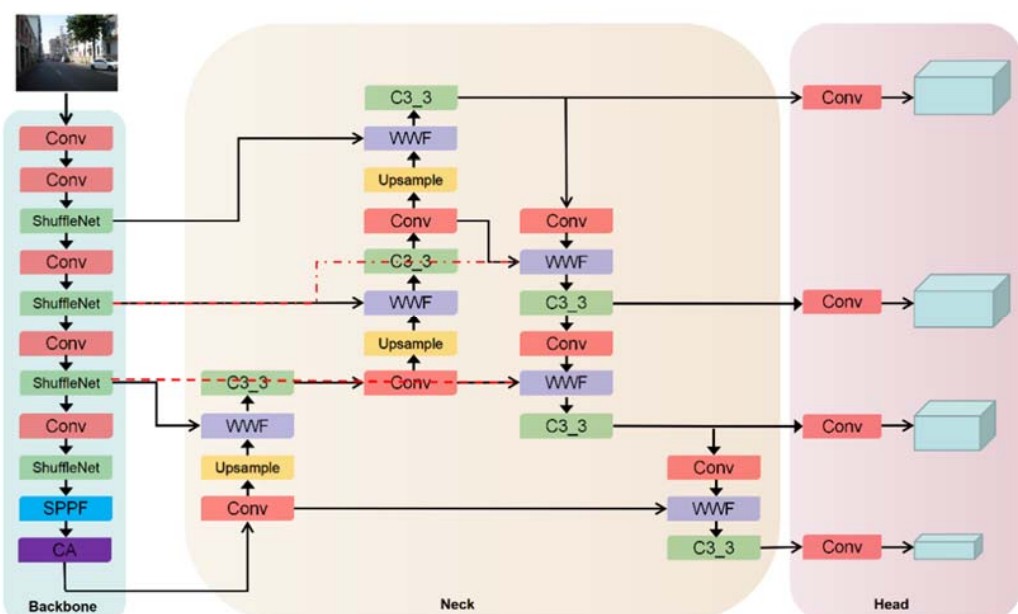

**Figure 2.** YOLO-FAM network.

### 2.1. ShuffleNet v2 Network Structure

The YOLOv5 initial model easily loses traffic sign feature information during the feature extraction process. The network model is large and has much parameter information.

The detection effect of traffic signs is low, and there are specific difficulties in the deployment process in this paper. ShuffleNet v2 [20] is used to replace the backbone network of YOLOv5, reducing parameters and realizing lightweight detection.

Figure 3 describes the ShuffleNet v2, which introduces the channel shuffle operation. While not increasing the amount of computation, the effect of feature extraction on traffic signs is enhanced. The ShuffleNet v2 network is divided into two units. In Unit 1, the feature channels are divided into two groups. To reduce the model's fragmentation, the network does nothing on the left side after performing a series of convolutions, BN, and Relu operations on the right side. Then, the network connects the output features of the left branch with the output features of the right and shuffles channels. In Unit 2, both the left and right components are downsampled, and a series of convolutions, BN, and Relu operations are carried out. Then, the network performs concreting and channel shuffling.

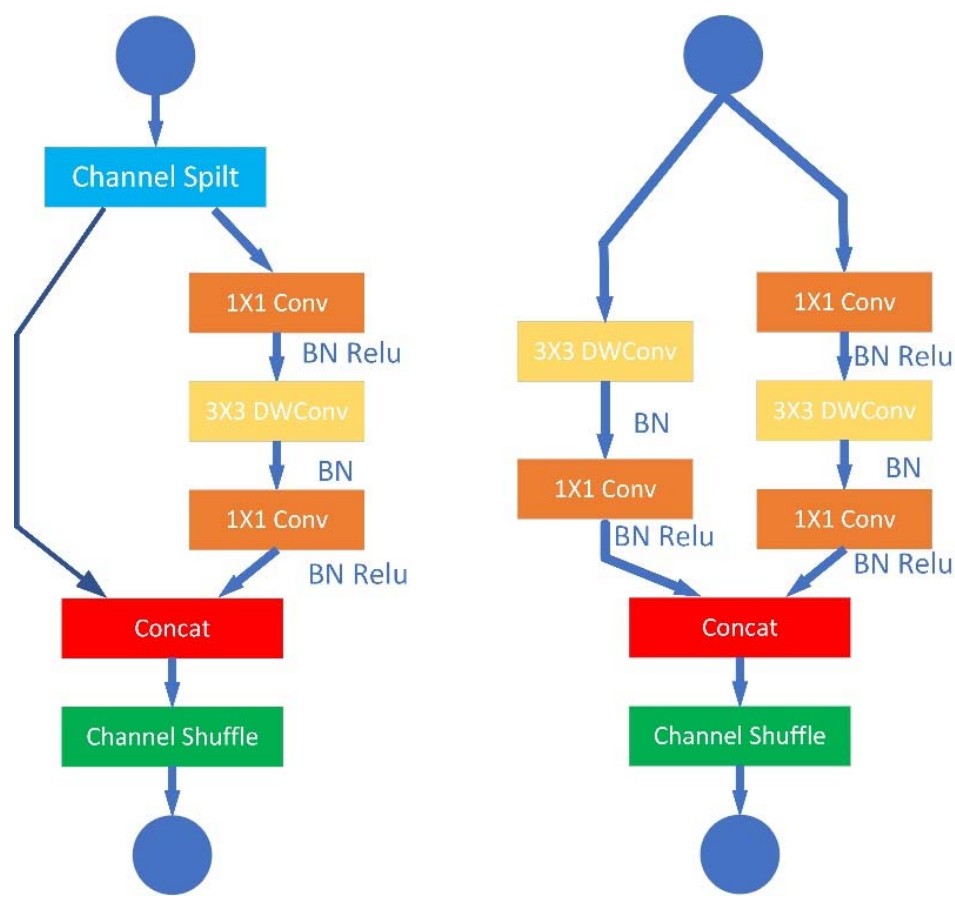

**Figure 3.** ShuffleNet v2 network structure.

### 2.2. Bi-FPN Network Structure

YOLOv5s adopts the PANet [21] structure for feature fusion. The PANet network introduces a bottom-up path, and low-level information is more easily passed on to the top of the high level. It then performs bottom-up feature fusion. However, to further strengthen the feature fusion method and obtain a better detection effect on traffic signs, we use the BIFPN network instead of the PANet network. The BiFPN network is shown in Figure 4. The BiFPN network enables simple and quick multi-scale feature fusion, adds an extra channel, integrates more features without increasing the cost, and obtains more feature information. BiFPN adopts a fast normalized fusion strategy. Each normalized weight takes a value between 0 and 1. The weighted fusion method is shown in Equation (1).

$$O = \sum_i \frac{\omega_i}{\varepsilon + \sum_j \omega_j} \times I_i \tag{1}$$

where $\varepsilon = 0.0001$ is added to the denominator to avoid numerical instability, $I_i$ is the input feature map, and $\omega_i$ and $\omega_j$ are the learnable weights of the input feature map, which can use the Relu activation function to ensure that its value is greater than 0.

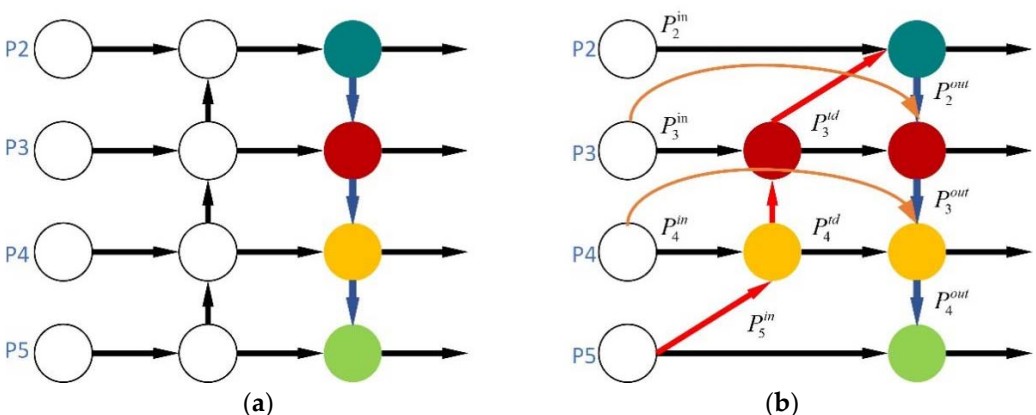

**Figure 4.** (**a**) PANet; (**b**) Bi-FPN.

The BiFPN network integrates bidirectional cross-scale connections and normalized fusion. As a specific example, this paper describes the feature fusion of BiFPN shown in Figure 4b at the P4 layer.

$$P_4^{td} = Conv\left(\frac{\omega_1 P_4^{in} + \omega_2 Resize\left(P_5^{in}\right)}{\omega_1 + \omega_2 + \varepsilon}\right) \tag{2}$$

$$P_4^{out} = Conv\left(\frac{\omega'_1 P_4^{in} + \omega'_2 P_4^{td} + \omega'_3 Resize\left(P_3^{out}\right)}{\omega'_1 + \omega'_2 + \omega'_3 + \varepsilon}\right) \tag{3}$$

where $P_4^{td}$ is the intermediate feature of the P4 layer, and $P_4^{out}$ is the output feature of the P4 layer. Re*size* is used for resolution matching of sampling operations. *Conv* is generally a convolution operation for feature processing.

### 2.3. CA Attention Mechanism

In detecting traffic signs, there is a problem of insufficient attention to the target in the salient area of the occluded target. Therefore, we add a CA attention mechanism to the feature fusion network. The CA attention mechanism is shown in Figure 5. It takes full advantage of the captured location information, captures the target area more accurately, and can effectively capture the relationship between channels. We encode horizontal and vertical position information into channel attention, conduct feature transformation by cascade fusion, and initiate $1 \times 1$ convolution Transform Function F. Then, we use two other $1 \times 1$ convolution transformation functions, $F_h$ and $F_w$, to output tensor through the sigmoid activation function. After feature integration, salient attention regions $y_c$ are obtained.

$$f = \beta\left(F\left(\left[z^h, z^\omega\right]\right)\right) \tag{4}$$

$$g^h = \delta\left(F_h\left(f^h\right)\right) \tag{5}$$

$$g^\omega = \delta(F_\omega(f^\omega)) \tag{6}$$

$$y_c(i, j) = x_c(i, j) \times g_c^h(i) \times g_c^\omega(j) \tag{7}$$

where $f$ is the feature map mapping, $\beta$ is the activation function, $z^h$, $z^\omega$ are the vertical and horizontal location information, $g^h$ and $g^\omega$ represent feature maps with the same quantity of output channels by sigmoid, and $x_c$ is feature information on skip connections.

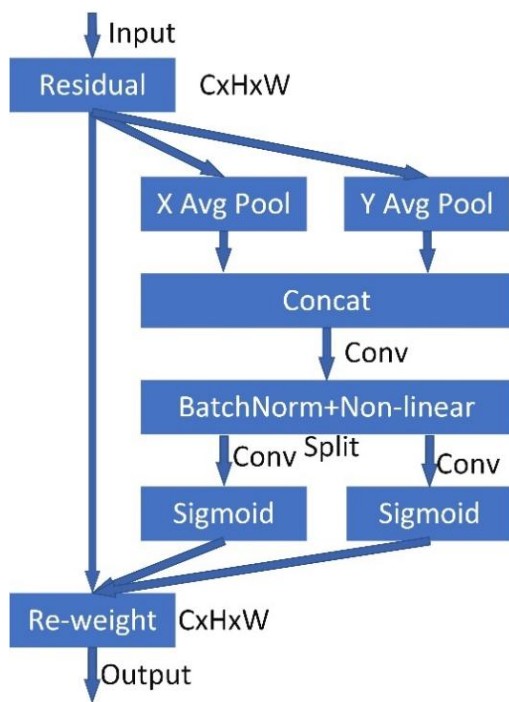

**Figure 5.** CA attention mechanism.

### 2.4. YOLOv5 Loss Function Improvement

The YOLOv5′s loss function consists of three parts: bounding box regression score, object score, and class probability score. In the bounding box regression score, complete intersection over union loss (CIOU Loss) is used to achieve prediction.

$$Loss = \lambda_1 L_{cls} + \lambda_2 L_{obj} + \lambda_3 L_{loc} \tag{8}$$

where $\lambda_1$, $\lambda_2$, $\lambda_3$ is the balance factor.

The loss function of YOLOv5 considers the overlapping area of bounding box regression, center point distance, and aspect ratio, but the formula reflects the difference in aspect ratio. As a result, the model cannot effectively optimize the similarity. For this problem, this paper adopts the better performance of the efficient intersection over union loss (EIOU Loss). Overlap loss, center distance loss, and width and height loss are the three components of the EIOU loss function. In the bounding box regression score, EIOU loss's width and height loss have a faster convergence speed and higher accuracy. It is better than the original network's CIOU loss.

$$L_{EIOU} = L_{IOU} + L_{dis} + L_{asp}$$
$$= 1 - IOU + \frac{\rho^2\left(b, b^{gt}\right)}{c^2} + \frac{\rho^2\left(\omega, \omega^{gt}\right)}{C_\omega^2} + \frac{\rho^2\left(h, h^{gt}\right)}{C_h^2} \tag{9}$$

where $c_\omega$ and $c_h$ are the width and height of the smallest bounding box covering the predicted and real boxes, $\rho$ is the Euclidean distance between $b$ and $b^{gt}$, $\omega$, $\omega^{gt}$ represent the width of the prediction box and the real box, respectively, and $h$, $h^{gt}$ represent the height of the prediction box and the real box, respectively.

### 3. Dataset and Experiment Setup

### 3.1. Image Dataset

This paper adopts the Chinese traffic sign database CTSDB [22]. As shown in Figure 6, China's traffic signs are divided into 64 categories, and data are divided into mandatory, prohibitory, and warning. The dataset includes realistic traffic scenarios recorded in various weather conditions. The total number of images used for training was 10,000. It should

be noted that for the case where some picture samples in the dataset are too few, pictures with less than 100 occurrences will be omitted. The images from the dataset are enhanced in this paper, as shown in Figure 7. In order to verify the effectiveness of the YOLO-FAM algorithm, the training set was 7000 and the test set was 3000.

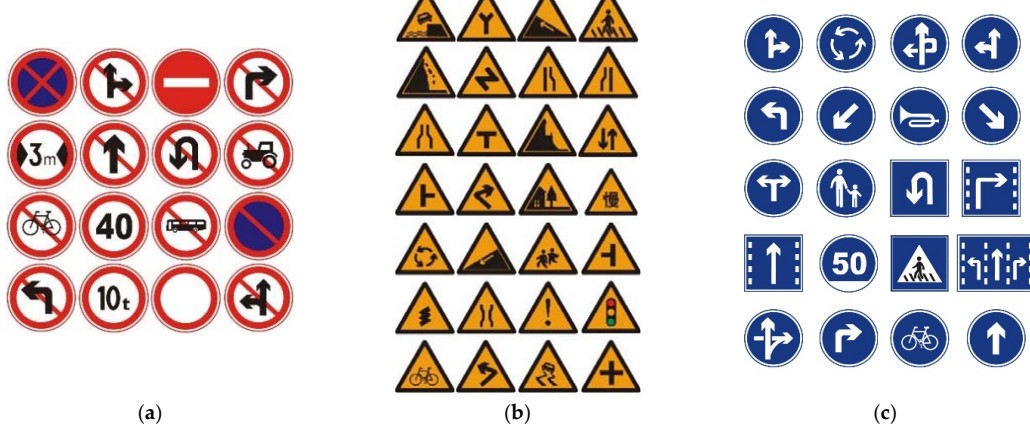

**Figure 6.** Chinese traffic signs: (**a**) prohibitory; (**b**) warning; (**c**) mandatory.

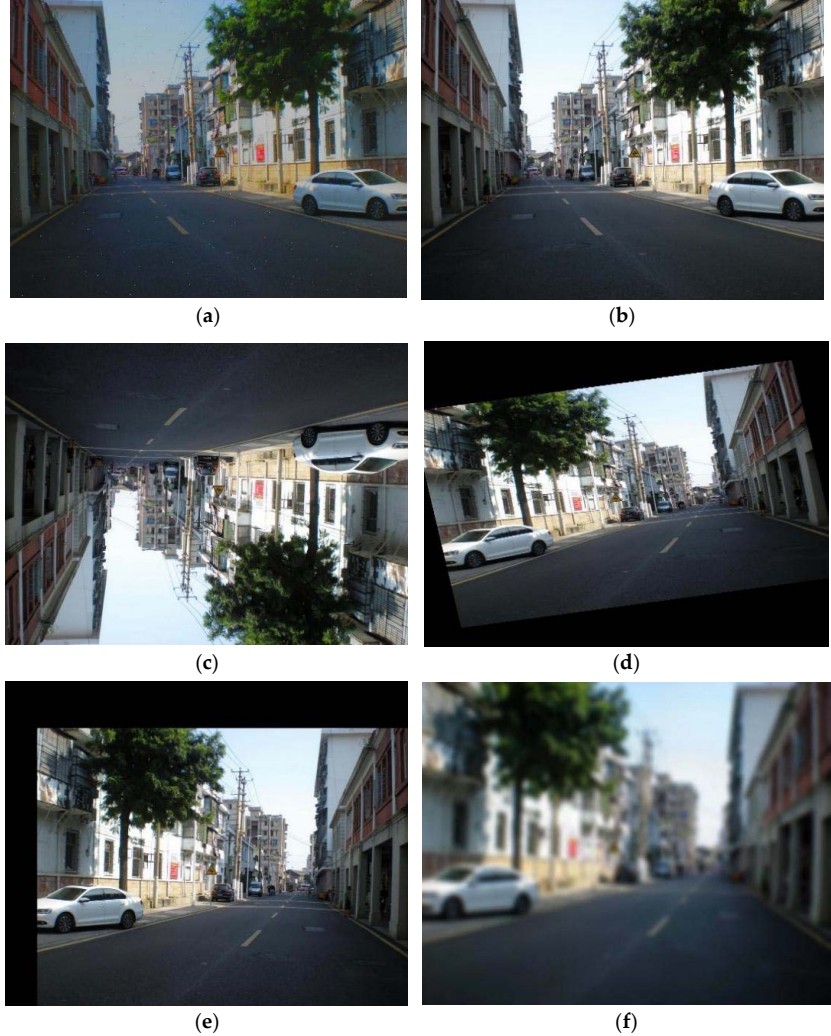

**Figure 7.** Image enhancement example: (**a**) add random noise; (**b**) horizontal flip; (**c**) vertical flip; (**d**) image rotation; (**e**) shift image; (**f**) image blurring.

## 3.2. Hardware Environment

In the experiment, Intel (R) Core (TM) i9-10700 CPU @ 3.70 GHz processor (Intel, Mountain View, CA, USA), 32 GRAM, and Nvidia GTX 2080 (NVIDIA, Santa Clara, CA, USA) were selected. All experiments were carried out in the environment of PyTorch 1.8, Cuda 10.0, Cudnn 7.6, Python 3.7, and Win 10.

## 3.3. Evaluation Indicators

In this paper, various types of experiments were carried out to verify and analyze actual performance. The purpose of this was to test the effectiveness of the proposed improved YOLO-FAM algorithm. We used several evaluation indicators to compare the performance of our method in both accuracy and real-time and compare it with other models with better performance.

The YOLO-FAM algorithm is mainly evaluated through parameters such as *Recall rate, Precision, mAP*, etc. FPS is the number of pictures processed per second. *TP* stands for true positives, *FN* stands for false negatives, and *FP* stands for false positives. Therefore, *Precision* and *Recall* were as follows:

$$Precision = \frac{TP}{TP + FP} \tag{10}$$

$$Recall = \frac{TP}{TP + FN} \tag{11}$$

$$AP = \int_{0}^{1} P(R) \tag{12}$$

$$mAP = \frac{1}{c} \sum_{j=1}^{c} AP_j \tag{13}$$

where *AP* stands for comprehensive evaluation of a particular category, *mAP* is obtained by averaging the mean precision (*AP*) across all classes, and c is a single class.

## 4. Experimental Results

### 4.1. Dataset Detection Results

To show the detection results of the YOLO-FAM algorithm for traffic signs, pictures of different environments were randomly selected from the CTSDB test set for detection.

This paper uses the YOLO-FAM algorithm, which uses multiple convolutional neural networks, to recognize traffic sign images. Then, various network structures were used to achieve feature fusion of different scales, and finally, target detection in different environments was realized.

Figure 8 depicts the experimental results under various environments. In comparison with the YOLO algorithm, the detection results were improved in multiple environments, proving that the YOLO-FAM algorithm recommended in this paper can easily complete the detection tasks in different environments.

### 4.2. Performance Comparison

To test the efficacy of the YOLO-FAM algorithm in traffic sign detection in this paper, the model was compared with the better performance models Faster-RCNN, YOLOv3, YOLOv4, YOLOv3-Tiny, YOLOv4-Tiny, and SSD algorithms. Table 1 shows the comparison. It can be seen from the table, as a large-scale network, Faster-RCNN has the advantage of high detection accuracy, with an average detection rate of 89.16%, but the high model complexity makes it difficult to deploy to mobile terminals with limited computing power. The accuracy of the improved YOLO-FAM model reaches 88.52%, which is only 0.64% behind compared with Faster-RCNN. The *mAP* values of YOLO-FAM increased by 21.31%

and 15.09%, respectively, compared with YOLOv3-Tiny and YOLOv4-Tiny.YOLO-FAM is more accurate and meets the real-time detection standard.

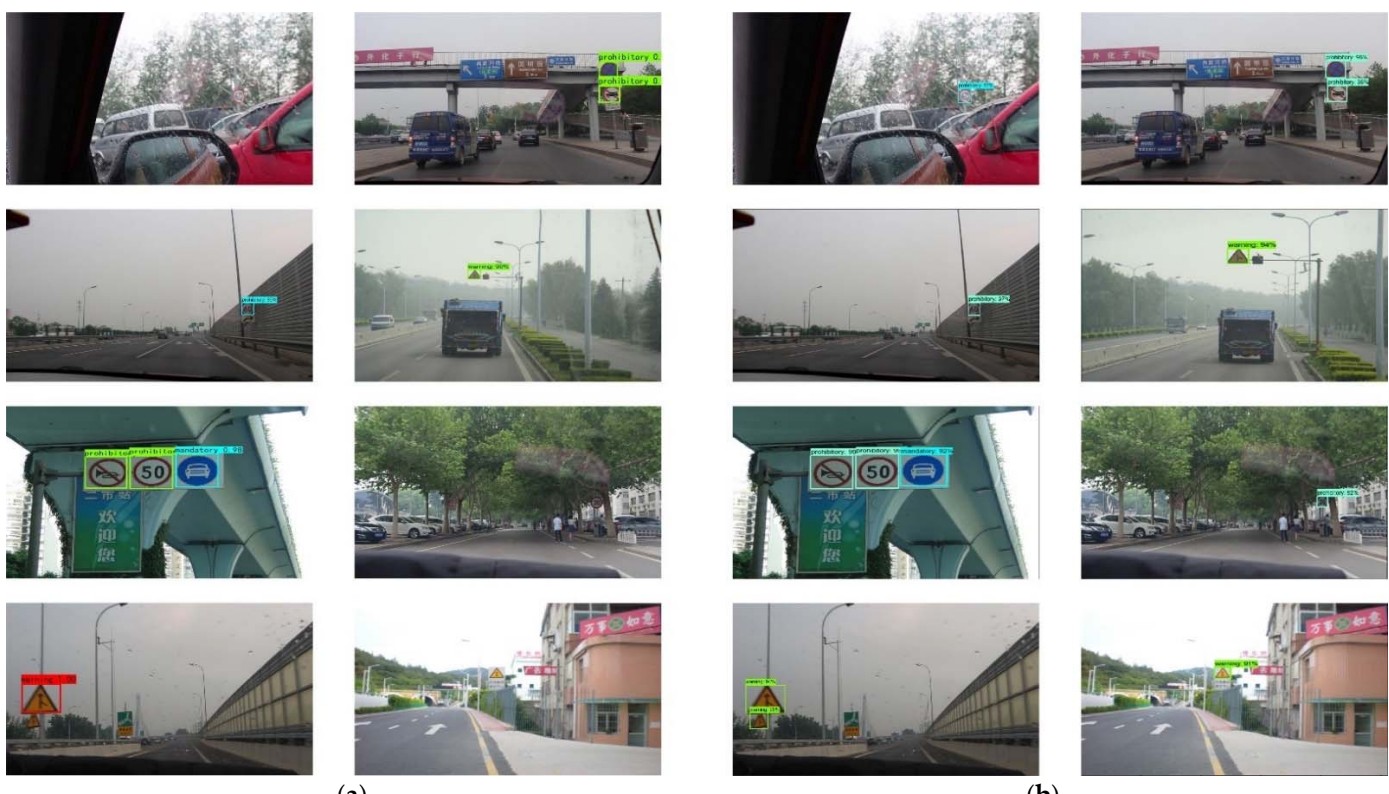

(**a**)                                           (**b**)

**Figure 8.** Comparison of recognition results of different detection frameworks: (**a**) YOLOv5; (**b**) YOLO-FAM.

**Table 1.** Experimental results comparison.

| Model | *mAP* | FPS | FLOPs (G) |
|---|---|---|---|
| Faster-RCNN | 89.16 | 17 | 535.7 |
| YOLOv3 | 83.76 | 29.6 | 66.5 |
| YOLOv4 | 88.24 | 41.2 | 60.2 |
| YOLOv5 | 86.25 | 94.2 | 9.5 |
| YOLOv3-Tiny | 67.21 | 79 | 6.1 |
| YOLOv4-Tiny | 73.43 | 95 | 6.9 |
| YOLO-FAM | 88.52 | 83.3 | 8.2 |

*4.3. Ablation Experiment*

The training performance evaluation was carried out based on the YOLOv5 algorithm, combined with different innovative strategies. The algorithm's recognition accuracy has been improved based on the assumption of ensuring real-time performance. Firstly, The ShuffleNet-v2 network replaced the c3 module. Since the c3 module has many parameters, the detection speed was slower. The ShuffleNet-v2 network is more convenient, which increases the detection speed of the algorithm, and the number of parameters is small. As shown in Table 2, *mAP* increased by 0.2%, FPS increased by 3.5%, and the number of parameters also decreased. Secondly, we joined the BIFPN network, introduced a simple and efficient weighted feature fusion mechanism, fused the effective information from the network's backbone, reduced interference to background information, and improved detection accuracy. Then, *mAP* was enhanced by 0.8%, and the identification speed also increased. Then, the CA attention mechanism was added. When detecting traffic signs, it

can identify channel information in the network structure and sense a range of direction and location information, helping YOLO-FAM locate and identify traffic sign information accurately. Finally, we used EIOU loss to accelerate convergence and improve regression accuracy. The four improved modules were added to the YOLOv5 algorithm, which increased the accuracy by 2.6%.

**Table 2.** Ablation Experiment Results.

| Methods | *mAP* (%) | FPS (f/s) | FLOPs (G) |
|---|---|---|---|
| YOLOv5 | 89.2 | 95.0 | 12.5 |
| YOLOv5 + ShuffleNet-v2 | 89.4 | 98.5 | 10.5 |
| YOLOv5 + ShuffleNet-v2 + BIFPN | 90.2 | 99.5 | 9.2 |
| YOLOv5 + ShuffleNet-v2 + BIFPN + CA | 92.4 | 100.1 | 8.5 |
| YOLOv5 + ShuffleNet-v2 + BIFPN + CA + EIOU | 92.5 | 95.5 | 8.9 |

*4.4. GTSDB Dataset Experimental Results*

To further test the improved YOLO-FAM model's detection effect on other traffic signs, the GTSDB [23] dataset was used for the experiments. Table 3 displays the experimental results. YOLOv5's anchor boxes were automatically learned from the training set, whereas YOLOv4's were not. Therefore, the identification accuracy and speed of YOLOv5 are superior to those of YOLOv4. YOLOv4-Tiny, although it adopts a lightweight backbone network, its target recognition accuracy is not very good. Compared with YOLOv5, the YOLO-FAM improves the backbone network and BIFPN network and locates traffic signs faster. The accuracy is significantly improved, as is the computational efficiency. Table 3 compares the detection results, and the YOLO-FAM algorithm achieves 87.82% *mAP* in the GTSDB dataset.

**Table 3.** Experiment results.

| Methods | *mAP* (%) | FPS (f/s) |
|---|---|---|
| YOLOv4 | 83.75 | 65.5 |
| YOLOv5 | 84.68 | 98.6 |
| YOLOv4-Tiny | 61.45 | 80.2 |
| YOLO-FAM | 87.82 | 89.2 |

**5. Conclusions**

In this paper, we proposed a YOLO-FAM method for traffic sign detection and applied it to the driving system. This method offers an improved image detection algorithm, YOLO-FAM, by improving the backbone network, adding the BiFPN network structure, adding an attention mechanism, and changing the loss function. Extensive experiments on the dataset show that the YOLO-FAM algorithm's *mAP* value in the CTSDB dataset is already improved by 2.27%. Finally, the YOLO-FAM algorithm's recognition results on the GTSDB dataset also have good results. The experiments demonstrate that the method YOLO-FAM can detect traffic signs effectively and quickly. We can apply the method proposed in this paper to the ADAS system, which can recognize traffic signs during driving. According to the information provided by the system, the driver can better make a series of responses to the traffic signs to better avoid traffic accidents.

Although the algorithm's accuracy has greatly improved, in general, the identification accuracy is still lower than that of large networks. There is still room for a significant improvement in practical applications. This article only classifies and recognizes traffic signs in China, but there are traffic signs that are not used in various countries at present, and the database needs to be improved to explore the classification and recognition of more types of traffic signs. At the same time, we also hope to develop a complete system for application in ADAS and better application in vehicles.

**Author Contributions:** Writing—original draft, H.Z. (Haohao Zou); writing—review and editing, H.Z. (Huawei Zhan); funding acquisition, L.Z. All authors have read and agreed to the published version of the manuscript.

**Funding:** This work is partially supported by the Hanan Provincial Natural Science Foundation Youth Science Fund Project (212300410185); Hanan Provincial University Key Research Project (21A510005).

**Institutional Review Board Statement:** Not applicable.

**Informed Consent Statement:** Not applicable.

**Data Availability Statement:** All results and data obtained can be found in open access publications.

**Conflicts of Interest:** The authors declare no conflict of interest.

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
