# Peer review of "Neural Network Based on Multi-Scale Saliency Fusion for Traffic Signs Detection"

_sustainability, doi:10.3390/su142416491_

Round 1
Reviewer 1 Report
This paper deals with automatic detection of road signs. I find this topic interesting and, to the best of my knowledge, the presented approach is novel. Methodology and experiments are OK, as well as the list of references. However, there are some minor things that need to be corrected.
1. Figure 1 is visually unclear for the reader.
2. Equations 10 and 11 suffer from minor spelling. Both Precision and Recall should be entirely written in italic.
3. Conclusion section should be extended.
4. There are three authors on the list. Authors stated "Author Contributions: Writing—original draft, H.Z.; writing—review and editing, H.Z.". Which H.Z is it? Zou or Zhan? What did Zhang do?
Reviewer 2 Report
Reviewer Comments
The authors have written on ‘Neural Network Based on Multi-Scale Saliency Fusion for Traffic Signs Detection”. I believe the readers would find the article more interesting if most of the sentences in the introduction were appropriately linked to each other. I suggest the service of a language editor to ensure that the manuscript is free from any grammatical errors. In addition to previously stated observations, the following suggestions should help improve the quality of the manuscript:
Abstract
1. Line 9 to line 10 is unclear; what do the authors mean by “traditional traffic object detection methods” the authors need to be clear on such words, especially for first-time readers.
2. Line, the authors said they proposed an improved network structure, or do they mean an improved neural network structure?
3. Firstly, use the ShuffleNet-v2 network to replace the c3 module in the YOLOv5 backbone network. Secondly, use the original PANet network of the BiFPN network, and add the CA attention mechanism. These lines are unclear and vague. Do the authors need to provide clarity why they are using such networks?
5. I am not convinced by this whole abstract. There is potential in the abstract. But many of the sentences in the abstract are too long and sometimes confusing. For example, where is the methodology in the abstract? A first-time reader does not know most of the abstract's acronyms. And the authors need to rewrite this abstract so that it can be as clear to a reader as possible. The abstract needs clarity and grammatical checks.
6. In the last lines of the abstract, the authors should indicate what is or are the significant contribution of their work to the field of road transportation, especially to road transportation, road intersections, specifically, to urban planners and transportation engineers.
7. The authors should rewrite the methodology in the abstract section. The methodology section in the abstract is not convincing enough to convince a first reader about what they used in this research.
Introduction
1. The first four lines of the abstract are unconvincing, and the sentences are vague and need to be written with clarity.
2. Most of the sentences in the introduction section are not grammatically correct and need a proper grammar check
3. The authors explain much about YOLO algorithm but fail to include related studies of where it has been used or applied in scholarly academic journals or conference papers.
4. The research contribution/aim/objectives or research questions are non-existent. The authors must create a subsection to state this clearly in the introduction section.
5. Research organization must also be stated as a subsection under the introduction section.
6. I am not convinced about the explanation of figure 1; first of all, this figure is too blurry, and the explanation of this figure needs to be more comprehensive and clearer.
7. The authors need to redo the introduction section and introduce recent literature and study background that can further support their research.
Methodology
1. A flowchart needs to be introduced at the beginning of the methodology section in order for readers to understand the conceptual and methodological framework of this research just by looking at the flowchart
2. Again, all figures are too blurry; these need to be corrected.
3. The authors need to introduce a step-by-step breakdown of why and how they were able to propose the YOLOv5- FAM algorithm in their research.
4. Line 183 to line 184 is confusing “The dataset includes natural traffic scenarios recorded in various weather conditions and many weather conditions” which one is it various weather conditions or many weather conditions?
5. How do the authors randomly select 10,000 images? The authors should explain this and not just put in a single line.
6. All acronyms must be clearly defined in a separate table and presented at the beginning or end of the paper.
7. The very important question is that the methodology puts more doubt on the traffic dataset used, i.e., data integrity.
Conclusion
1. The conclusion needs to be re-written. The authors failed to include any recommendations or future works in the conclusion, nor do they tell the readers the implication of their model, contributions of their model, why researchers or transportation engineers should apply this method, is the methodology practical enough to be used in real-life traffic flow situations.
2. The first four lines of the conclusion section look like the research contribution or research aim, this needs to be corrected and the conclusion rewritten.
Author Response
请参阅附件

Reviewer 3 Report
This article is about automatic traffic sign detection systems. These systems play a fundamental role for the circulation of automatic detection vehicles, since they are essential for a circulation without mishaps, specially in unmanned vehicles. Therefore, precise methods are needed that are capable of identifying this type of signals in moving images despite the fact that the weather, traffic, or the environment itself make it difficult. The article presented by the authors is based on research on the improvement of elements that have been used for the detection of traffic signs, obtaining good results. The article is suitable for publication, however, there are some details to improve, which are listed below.
Line 50. “thus” à “Thus”
Figure 1. Please, improve the quality of the figure, as many words are imposible to read. Also the size of the text is very small; I suggest to make the text larger
Line 126: “For“ à “for”
Line 129: “The“ à “the”
Lines 134—135: Pleases, review the meaning of the sentences, and the punctuation marks
Reviewer 4 Report
This paper proposes an improved image detection algorithm YOLO-FAM by improving the backbone network, adding BiFPN network structure, adding an attention mechanism, and changing loss function. The experimental results show the effectiveness of the improved algorithm.
(1) Figure 7 does not intuitively illustrate the performance of the proposed algorithm. Comparative results should be displayed. The reviewer suggested that the author discuss different environments separately to illustrate the performance of the algorithm proposed in this paper in complex environments.
(2) The author needs to be clear about the training time of each algorithm
Round 2
Reviewer 2 Report
Dear Authors,
I am satisfied with your answer to my comments.